chemical biology/medicinal chemistry

N^ε-acetyl lysine, ZBG group, analogues, hybridization, HDAC inhibitor

**Author for correspondence:**
Bin He
e-mail: binhe@gmc.edu.cn

# N^ε-acetyl lysine derivatives with zinc binding groups as novel HDAC inhibitors

Fang Wang[1,3], Chun Wang[1,3], Jie Wang[1,3], Yefang Zou[1,3], Xiaoxue Chen[1,3], Ting Liu[2,3], Yan Li[1,4], Yonglong Zhao[1,3], Yongjun Li[1,3] and Bin He[1,3]

[1]State Key Laboratory of Functions and Applications of Medicinal Plants, Engineering Research Center for the Development and Application of Ethnic Medicine and TCM (Ministry of Education), [2]Guizhou Provincial Key Laboratory of Pharmaceutics, [3]School of Pharmacy, and [4]School of Basic Medicine, Guizhou Medical University, Guiyang 550004, People's Republic of China

BH, 0000-0001-6884-6166

HDAC inhibitors have been developed very rapidly in clinical trials and even in approvals for treating several cancers. However, there are few reported HDAC inhibitors designed from N^ε-acetyl lysine. In the current study, we raised a novel design, which concerns N^ε-acetyl lysine derivatives containing amide acetyl groups with the hybridization of ZBG groups as novel HDAC inhibitors.

## 1. Introduction

Histone deacetylases (HDACs) are a class of hydrolases that remove acetyl groups from lysine residues of proteins, and play a very important role in the regulation of many biological processes, including transcription, genome stability, metabolism, protein activity, lifespan and so on [1–4]. According to sequence identity and similarity, human HDACs have been typically divided into four classes [5,6]. Class I consists of HDAC 1, 2, 3 and 8 while Class II includes HDAC 4, 5, 6, 7, 9 and 10, which is further divided into two subclasses: Class IIa (HDAC 4, 5, 7 and 9) and Class IIb (HDAC 6 and 10). Class IV has only one member, called HDAC 11. Notably, Class I, Class II and Class IV are all Zinc ($Zn^{2+}$) dependent deacetylases (figure 1a), whereas Class III is a family of nicotinamide adenine dinucleotide (NAD)-dependent deacylases, which is also known as sirtuin and contains seven members (SIRT 1–7). Because of their critical role in cell proliferation, cell cycle and apoptosis of cancer cells, HDACs have been considered as promising therapeutic targets for treating cancer [7–13]. Furthermore, the development of HDAC inhibitors has been proven to be an efficient strategy for cancer treatment.

**Figure 1.** HDAC ($Zn^{2+}$-dependent deacetylases) and approved HDAC inhibitors. (*a*) HDAC catalysed deacetylation; (*b*) the structural features of approved HDAC inhibitors: their common structural characteristics have been defined as three components, which are a cap group as surface recognition marked with green, a linker marked with black, and a zinc binding group (ZBG) marked with red that can chelate the zinc (II) cation.

Indeed, there are many HDAC inhibitors currently in clinical trials and there are even five HDAC inhibitors already on the market. Vorinostat (SAHA) **1** [14], belinostat (PXD-101) **2** [15] and romidepsin (FK228) **3** [16] have been approved by the US Food and Drug Administration (FDA) for treating cutaneous T-cell lymphoma (CTCL) or peripheral T-cell lymphoma (PTCL) while panbinostat (LBH-589) **4** [17] has also been approved by the FDA for the treatment of multiple myeloma (figure 1*b*). Recently, chidamide **5** [18] was approved by the China Food and Drug Administration for the treatment of PTCL (figure 1*b*).

Of these five approved HDAC inhibitors, vorinostat (SAHA) **1**, belinostat (PXD-101) **2**, romidepsin (FK228) **3** and panbinostat (LBH-589) **4** are all pan-HDAC inhibitors, which exhibit a lack of isoform selectivity (figure 1*b*) [14–17]. However, chidamide **5** is the first selective HDAC inhibitor to obtain marketing approval in China so far (figure 1*b*) [18]. Whether they are pan- or selective- HDAC inhibitors, three common structural characteristics have been defined, which are a cap group as surface recognition (marked with green), a linker (marked with black) and a zinc binding group (ZBG) (marked with red) that can chelate the zinc (II) cation (figure 1*b*). Typically, ZBG groups include the hydroxamate group, thiol group and amino benzamide.

Although considerable progress has been made in the development of HDAC inhibitors, clinically used HDAC inhibitors still have some side effects, such as excessive toxicities, instability and off-target effects [19–23]. Therefore, the continued development of novel HDAC inhibitors is needed to avoid side effects and improve their pharmacological and pharmacokinetic properties [24–26]. At present, there are few HDAC inhibitors designed from $N^{\varepsilon}$-acetyl lysine (HDAC substrate) while the

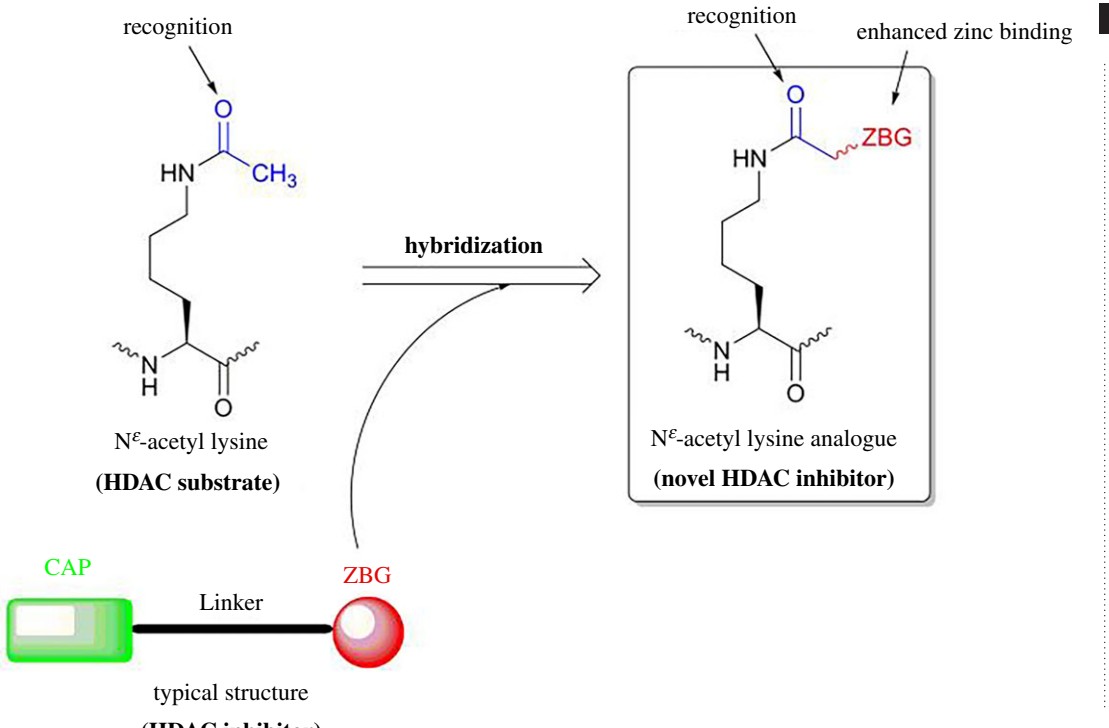

**Figure 2.** The design of a novel HDAC inhibitor based on N$^\varepsilon$-acetyl lysine and ZBG of typical HDAC inhibitors.

designs of sirtuin inhibitors by mimicking N$^\varepsilon$-acyl lysine (sirtuin substrate) have led to many successful examples [27–30]. Considering this, our design herein is based on the hypothesis that N$^\varepsilon$-acetyl lysine (HDAC substrate) derivatives containing amide acetyl groups with the hybridization of the ZBG group may help their recognition of HDAC and further enhance the zinc binding in the HDAC active site, thus inhibiting HDAC activity (figure 2). We report here the synthesis of the new hybrid compounds, the evaluation of their HDAC inhibition and preliminary results in anti-cancer activities on several cancer cell lines.

# 2. Results and discussion

## 2.1. Chemistry

The synthesis of N$^\varepsilon$-acetyl lysine derivatives started from the condensation of commercial N$^\varepsilon$-tert-Butyloxycarbonyl(Boc)-N$^\alpha$-carbobenzoxyl(Cbz)-L-lysine **6** with aniline in the presence of N,N'-dicyclohexylcarbodiimide as a coupling reagent to obtain compound **7**. After the deprotection of the Boc group by treating compound **7** with trifluoroacetic acid, the key intermediate **8** was achieved in a yield of 85% for two steps. After the reaction of compound **8** with succinic anhydride or maleic anhydride, compounds **9** and **10** were obtained, respectively. The desired hydroxamic acid **11** was achieved by the treatment of compound **9** with hydroxyl amine while the amino benzamide **12** was given by the coupling of 1,2-diaminobenzene with compound **9**. Treatment of compound **10** with hydroxyl amine gave another desired hydroxamic acid **13** (scheme 1).

The intermediate **8** was coupled with different heterocyclic acids or an aromatic acid (1H-indene-3-carboxylic acid) to give compounds **14a-f**, **14h-k** and **14m-p**, respectively. Compound **14g** containing the thiol group was achieved by the condensation of compound **8** and 2-mercaptoacetic acid. The condensation of the intermediate **8** with indole-3-carboxaldehyde and then the reduction of imine gave compound **14l**. Additionally, the deprotection of the Boc group of compound **14p** by TFA gave compound **14q** (Scheme 2).

Amino benzamides **17a-c** were synthesized from the intermediate **8**. The intermediate **8** was first coupled with mono ethyl or monomethyl $\alpha$, $\omega$-dicarboxylic esters to give compounds **15a-c**, which was hydrolysed to give compounds **16a-c**. Then, the intermediates **16a-c** were coupled with 1,2-diaminobenzene to obtain amino benzamides **17a-c**, respectively. On the other hand, by the

**Scheme 1.** First run synthesis of the key intermediate **8** and compounds **9–13**. Reagents and conditions: (a) aniline, 2-(1H-benzotriazole-1-yl)-1,1,3,3-tetramethyluronium hexafluorophosphate (HBTU), N,N-diisopropylethylamine (DIEA), tetrahydrofuran (THF), 3 h, 90%; (b) 33% trifluoroacetic acid (TFA) in CH₂Cl₂, 1 h, 95%; (c) succinic anhydride, triethylamine (TEA), THF, rt, 1 h, 90%; (d) maleic anhydride, TEA, rt, 1 h, 90%; (e) (i) isobutyl chloroformate (IBCF), TEA, THF, (ii) NH₂OH·HCl, MeOH, 0°C to rt, 3 h, 42%; (f) 1,2-diaminobenzene, HBTU, DIEA, 80%, 4 h; (g) (i) IBCF, TEA, THF, (ii) NH₂OH·HCl, MeOH, 0°C to rt, 3 h, 40%.

**Scheme 2.** Second run synthesis of compounds **14a-q**. Reagents and conditions: (a) HBTU, DIEA, indicated heterocyclic acids or an aromatic acid (1H-indene-3-carboxylic acid), rt, 3 h, 20–85% for **14a-k** and **14m-q**; (b) (i) indole-3-carboxaldehyde, dry MeOH, (ii) NaBH₄, MeOH, 78% for **14l**; (c) 33% trifluoroacetic acid (TFA) in CH₂Cl₂, 1 h, 95%.

treatment of intermediates **16b-c** with hydroxyl amine, hydroxamic acids **18b-c** were finally obtained (scheme 3).

## 2.2. HDAC inhibition, cellular study and antiproliferative activity

With these Nᵉ-acetyl lysine derivatives in hand (schemes 1–3), we then did the pilot screening for general HDAC inhibitory activity at a compound's concentration of 100 µM. All tested compounds were subjected to the inhibition assay against the HDAC deacetylation reaction by using a HeLa nuclear

**Scheme 3.** Third run synthesis of amino benzamides **17a-c** and hydroxamic acids **18b-c**. Reagents and conditions: (a) indicated monoethyl or monomethyl $\alpha,\omega$-dicarboxylic esters, HBTU, DIEA, THF, rt, 3 h, 73−88% for **15a-d**; (b) LiOH, 25% $H_2O$ in THF, 0°C, 0.5 h, approximately 90%; (c) 1,2-diaminobenzene, HBTU, DIEA, rt, THF, 3 h, 60−80% for **17a-c**; (d) $NH_2OH \cdot HCl$, MeOH, rt, 40−60% for **18b-c**.

**Table 1.** Pilot screening of HDAC inhibiton for $N^{\varepsilon}$-acetyl lysine derivatives containing acetyl group with the hybridization of ZBG groups (*100 $\mu$M; **10 $\mu$M).

| *Cmpd | HDAC inhibition (%) | *Cmpd | HDAC inhibition (%) |
|---|---|---|---|
| **9** | 37.5 ± 0.6 | **14m** | 14.3 ± 0.5 |
| **10** | 38.7 ± 1.0 | **14n** | 16.2 ± 2.0 |
| **11** | 86.0 ± 3.0 | **14o** | 48.6 ± 0.5 |
| **12** | 30.7 ± 3.7 | **14p** | 24.7 ± 2.8 |
| **13** | 39.3 ± 3.1 | **14q** | 36.2 ± 0.4 |
| **14a** | 39.6 ± 0.4 | **15a** | 19.4 ± 10.9 |
| **14b** | 39.2 ± 0.8 | **15b** | 2.9 ± 0.8 |
| **14c** | 51.0 ± 1.0 | **15c** | 7.1 ± 0.3 |
| **14d** | 48.4 ± 4.4 | **16a** | 4.0 ± 1.7 |
| **14e** | 54.0 ± 1.9 | **16b** | 4.3 ± 0.5 |
| **14f** | 45.0 ± 0.4 | **16c** | 1.1 ± 0.4 |
| **14g** | 10.3 ± 4.9 | **17a** | 7.0 ± 0.4 |
| **l4h** | 28.3 ± 4.6 | **17b** | 69.3 ± 3.4 |
| **14i** | 36.8 ± 1.1 | **17c** | 50.6 ± 6.9 |
| **14j** | 12.2 ± 6.5 | **18b** | 99.9 ± 0.9 |
| **14k** | 22.4 ± 0.2 | **18c** | 109.2 ± 0.5 |
| **14l** | 3.1 ± 0.5 | ****SAHA (1)** | 102.2 ± 4.1 |

extract as a source of HDACs and BOC-Ac-Lys-AMC as a substrate. As shown in table 1, hydroxamic acid **11** showed better HDAC inhibition ($86.0 \pm 3.0\%$) than that of other first-run synthesized compounds (**9, 10, 12, 13**) (scheme 1). This indicated that the hybridization of hydroxamic acid might be the best choice compared to the hybridization of other ZBG groups like acid or amino benzamide. In the second run synthesis, we have incorporated not only another classic ZBG group like the thiol group but also other potential ZBG groups like heterocyclic groups into the candidate compounds (scheme 2). Unfortunately, none of them (**14a-q**) showed superior inhibition compared to that of

**Table 2.** IC$_{50}$ values ($\mu$M) of N$^{\varepsilon}$-acetyl lysine derivatives as HDAC inhibitors.

| inhibitor | HeLa nuclear extract |
| --- | --- |
| **11** | 25.36 $\pm$ 1.35 |
| **18b** | 10.44 $\pm$ 3.86 |
| **18c** | 0.50 $\pm$ 0.21 |
| **SAHA (1)** | 0.05 $\pm$ 0.01 |

hydroxamic acid **11** (**14a-q** versus **11**, table 1). Although N$^{\varepsilon}$-acetyl lysine derivatives with the hybridization of different heterocyclic groups could not greatly improve the inhibitory potency compared with those with the hybridization of aromatic group or non-keto heterocyclic group (**14a-f**, **14h-k** versus **14k** or **14l**, table 1), most of those with the hybridization of heterocyclic groups did show some degree of HDAC inhibition (table 1). Among them, the best one is **14e** with a furan group as a ZBG group showed the HDAC inhibition of 54.0 $\pm$ 1.9% at 100 $\mu$M (table 1). The introduction of the benzo group into the six-member heterocyclic group showed no obvious effects on the HDAC inhibition (**14h-i** versus **14a-b**, table 1). However, the HDAC inhibition of **14m** was dropped to 14.3 $\pm$ 0.5% when the furan group was replaced by the 2,3-benzofuran group (**14m** versus **14e**, table 1). Similarly, the HDAC inhibitions of **14j** and **14n** were both decreased to 12.2 $\pm$ 6.5% and 16.2 $\pm$ 2.0%, respectively (**14j** versus **14d** and **14n** versus **14f**, table 1).

To further improve the inhibitory potency, we performed the third run synthesis of amino benzamides **17a-c** and hydroxamic acids **18b-c**, and their evaluation for HDAC inhibition. Again, amino benzamide and hydroxamic acid have been confirmed to be the most appropriate ZBG groups compared to carboxyl acid or carboxyl ester (**17a-c** and **18b-c** versus **15a-c** and **16a-c**, table 1). By optimizing the length between the carbonyl of amide acetyl moiety and amino benzamide as a ZBG group, we found that amino benzamides **17b** and **17c** gave the better HDAC inhibition of 69.3 $\pm$ 3.4% and 50.6 $\pm$ 6.9%, respectively (**17b-c** versus **17a**, table 1). Finally, hydroxamic acids **18b** and **18c** demonstrated the best inhibitory potency of 99.9 $\pm$ 0.9% and 109.2 $\pm$ 0.5% at 100 $\mu$M, respectively, which is comparable to the HDAC inhibition of SAHA (**1**) at 10 $\mu$M. Additionally, **18c** and SAHA (**1**) demonstrated no obviously inhibitory selectivity between HDAC I and HDAC II (electronic supplementary material, table S1).

To further evaluate the inhibitory potency of hydroxamic acids **11** and **18b-c**, we measured their IC$_{50}$ values through the HDAC deacetylation reaction by using a HeLa nuclear extract as a source of HDACs and BOC-Ac-Lys-AMC as a substrate (table 2). The IC$_{50}$ values of hydroxamic acids **11** and **18b** were 25.36 $\pm$ 1.35 $\mu$M and 10.44 $\pm$ 3.86 $\mu$M, respectively (table 2). Hydroxamic acid **18c** eventually turned out to be the most potent HDAC inhibitor with a IC$_{50}$ value of 0.50 $\pm$ 0.21 $\mu$M, which like SAHA fell into the nanomolar range with an IC$_{50}$ value of 0.05 $\pm$ 0.01 $\mu$M (**18c** versus SAHA (**1**), table 2).

This result encouraged us to further evaluate whether hydroxamic acid **18c** could work on the cellular HDACs. Therefore, hydroxamic acid **18c** was engaged in western blotting analysis to evaluate the acetylation levels of histone H3 and $\alpha$-tubulin. The leukaemia K562 and human non-small cell lung cancer A549 cell lines were treated with hydroxamic acid **18c** at 50, 100 and 200 $\mu$M for 24 h, in comparison with SAHA (**1**) at 10 $\mu$M as a positive control. As shown in figure 3, hydroxamic acid **18c** induced a dose-dependent increase in acetylation levels of histone H3 and $\alpha$-tubulin in both K562 and A549 cell lines. In other words, hydroxamic acid **18c** was able to increase the acetylation levels of histone H3 and $\alpha$-tubulin, suggesting that **18c** could inhibit HDACs in cells.

Because of the great anti-tumour potential of HDAC inhibitors, we did the antiproliferative evaluation of **18c** against K562 cells, A549 cells and HepG2 cells using MTS assay as previously described [7–13]. Hydroxamic acid **18c** displayed the antiproliferative activities in tested cancer cell lines in a dose-dependent manner within a concentration range of 10–500 $\mu$M (figure 4$a$–$c$). In K562 and A549 cells, **18c** at 50–100 $\mu$M showed comparable antiproliferative activity to that of SAHA (**1**) at 10 $\mu$M (figure 4$a$) while **18c** at 200 $\mu$M demonstrated comparable antiproliferative activity to that of SAHA (**1**) at 10 $\mu$M in HepG2 cells (figure 4$b$–$c$) (electronic supplementary material, table S2). Additionally, **16c** as a negative control compound with 500 $\mu$M showed no obvious antiproliferative activities in all tested tumour cell lines (figure 4$a$–$c$), indicating the cytotoxicity of **18c** was contributed by its HDAC inhibition. More importantly, like the negative control compound **16c**, hydroxamic acid **18c** showed less toxicity in one non-cancerous kidney cell (HEK293) than that of SAHA (**1**) at the same concentrations (5 $\mu$M and 10 $\mu$M,

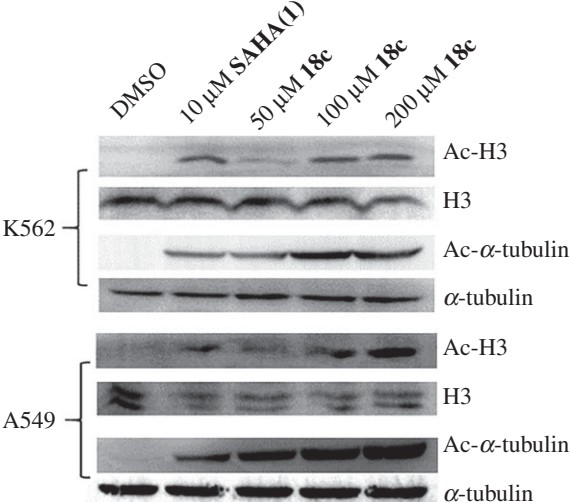

**Figure 3.** Cellular HDAC inhibition of hydroxamic acid **18c** by western blotting analysis of the acetylation levels of histone H3 and α-tubulin in K562 and A549 cell lines.

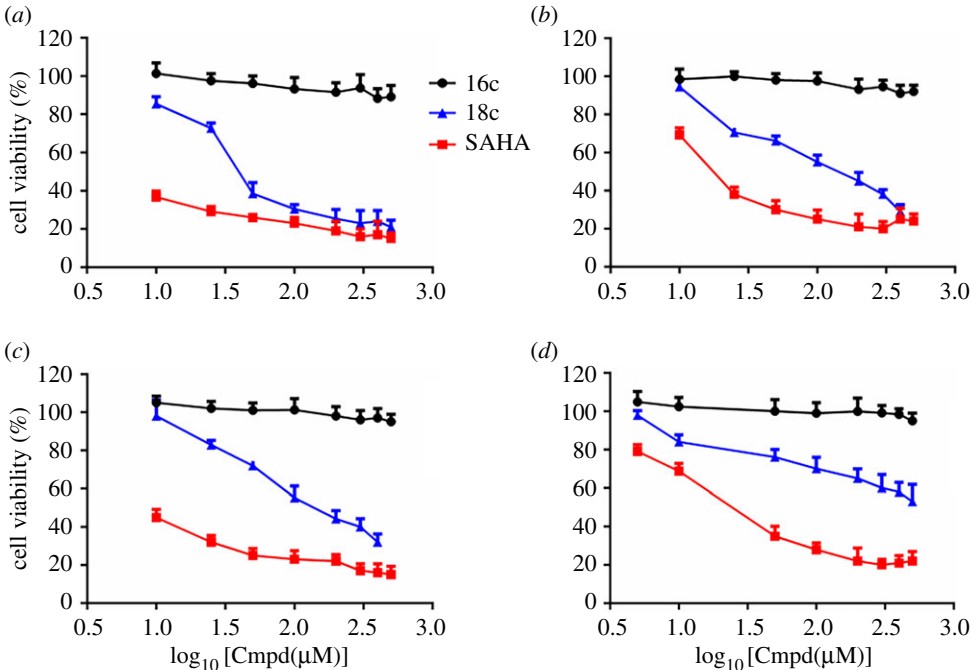

**Figure 4.** The antiproliferative evaluation of **18c** against K562 cells (*a*), A549 cells (*b*), HepG2 cells (*c*) and HEK293 (*d*) using MTS assay. (Cmpd: tested compound **16c**, **18c** or **SAHA**).

figure 4*d* ), which demonstrated that **18c** may have the therapeutic potential for targeting cancer cells but less toxicity for normal cells (electronic supplementary material, table S2).

# 3. Conclusion

Using Nᵉ-acetyl lysine (HDAC substrate), we raised a novel design, concerning Nᵉ-acetyl lysine derivatives containing amide acetyl groups with the hybridization of ZBG groups as novel HDAC inhibitors. This idea is triggered by our successful design of sirtuin inhibitors mimicking Nᵉ-acyl lysine [31–34]. Then, we synthesized 33 small molecules as candidates by using acetyl lysine successively hybridized with carboxylic ester or acid, hydroxamic acid, amino benzamide, thiol and heterocyclic groups as ZBG groups. After evaluation of these compounds, we found that the

compounds **11** and **18b-c** hybridized with hydroxamic acid demonstrated superior HDAC inhibition in comparison with all tested compounds. The best one is compound **18c**, with an $IC_{50}$ value of approximately 500 nM, which also can inhibit cellular HDACs (table 2 and figure 3). Most importantly, **18c** in a concentration range of 50 to 200 μM showed comparable antiproliferative activity to that of SAHA (**1**) at 10 μM in all tested human tumour cell lines (K562, A549 and HepG2) while **18c** showed less toxicity in one non-cancerous kidney cell (HEK293) than that of SAHA (**1**) at the same concentrations (figure 4 and electronic supplementary material, table S2). The inhibitory mechanism is possibly that the unit of amide acetyl lysine in **18c** may help its recognition of HDAC and the unit of hydroxamic acid in **18c** further enhances the zinc binding in the HDAC active site, and thus inhibits HDAC activity (figure 2). Eventually, this result tells us that the novel design of acetyl lysine with the hybridization of ZBG groups has opened up a new direction and could be exploited for developing more therapeutic HDAC inhibitors. The study of HDAC inhibitor **18c** as a leading compound for medicinal chemistry is underway in our laboratory.

Data accessibility. Data available from the Dryad Digital Repository: https://doi.org/10.5061/dryad.d1t59pp [35].

Authors' contributions. F.W. carried out most synthetic and biological work, and participated in data analysis; C.W. and J.W. participated in synthetic work; Y.Z. and X.C. participated in biological work in biological work; T.L., Yan L. and Yongjun L. participated in the data analysis, discussion and proofreading; B.H. did the design and drafted the manuscript. All authors gave final approval for publication.

Competing interests. We have no competing interests.

Funding. This study was supported by National Natural Science Foundation of China (grant no. 21662010), Guiyang Municipal Science and Technology Project (grant no. [2017] 30-18/30-38), Department of Science and Technology of Guizhou Province (grant nos. [2019]2760, [2018]5779-68 and [2016]5613/5677), One hundred Talents Program of Guizhou Province (2015), the Engineering Center of Natural Science Foundation of Department of Education of Guizhou Province (Microbiology and Biochemical Pharmacy, [2015]338).

Acknowledgements. We thank Prof. Hening Lin from Cornell University and Dr Yi Wang from Hong Kong University give us helpful suggestions during the research.

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
