## [Reviewer comments · Royal Society Open Science]

Review History

RSOS-190338.R0 (Original submission)

Review form: Reviewer 1

Is the manuscript scientifically sound in its present form?

Yes

Are the interpretations and conclusions justified by the results?

Yes

Is the language acceptable?

Yes

Is it clear how to access all supporting data?

Yes

Do you have any ethical concerns with this paper?

No

Have you any concerns about statistical analyses in this paper?

No

Recommendation?

Accept with minor revision (please list in comments)

Comments to the Author(s)

HDAC is an important epigenetic target in drug discovery and development. The authors developed N ϵ -acetyl-lysine analogues with Zinc binding groups as HDAC inhibitors. The best compound 18c had an IC₅₀ value of 500 nM for HDACs. Besides, WB and MTS assay were used to evaluate the antitumor activity of all synthetic compounds. This manuscript has provided an effective strategy for developing novel HDAC inhibitors.

Minor comments:

- 1) the values of the anti-proliferative assay in Fig. 4 should be provided based on dose-dependent curve-fitting.
- 2) in order to evaluate 18c toxicity, the authors need to determine the IC₅₀s of 18c and SAHA against both tumor cells and normal cells, then calculate and compare their selective index.

Review form: Reviewer 2

Is the manuscript scientifically sound in its present form?

Yes

Are the interpretations and conclusions justified by the results?

Yes

Is the language acceptable?

Yes

Is it clear how to access all supporting data?

Yes

Do you have any ethical concerns with this paper?

No

Have you any concerns about statistical analyses in this paper?

I do not feel qualified to assess the statistics

Recommendation?

Accept with minor revision (please list in comments)

Comments to the Author(s)

This manuscript described the design and synthesis of HDAC inhibitors by hybridizing acetyl-lysine and zinc binding group. Their enzymatic and cellular activities were also characterized. The most promising compound 18c had reasonable inhibition against HDACs. Interestingly, 18c demonstrate the comparable antiproliferative activities against several cancer cell lines but the less toxicity for normal cells than that of SAHA. Therefore, it is worthy of eventual publication in Royal Society Open Science if the authors can address the following concerns:

1. Should add some discussion about whether the synthesized compounds intend to be pan- or specific- HDAC inhibitors.

2. The cellular data is just described for the acetylation level of tubulin which is only relevant for HDAC6. HDAC1-3 are involved in acetylation of H3, which is strongly suggested to be present in the figure.
3. The authors should give dose-dependent curves in Fig. 4.
4. The authors should check English and careless typos in the text, and the format of the structural formula in SI.

Review form: Reviewer 3

Is the manuscript scientifically sound in its present form?

Yes

Are the interpretations and conclusions justified by the results?

No

Is the language acceptable?

No

Is it clear how to access all supporting data?

Yes

Do you have any ethical concerns with this paper?

No

Have you any concerns about statistical analyses in this paper?

I do not feel qualified to assess the statistics

Recommendation?

Major revision is needed (please make suggestions in comments)

Comments to the Author(s)

N ϵ -Acetyl-lysine analogs with Zinc binding groups as novel HDAC inhibitors
Fang Wang, et al

The manuscript describes the design, synthesis, and evaluation of a series of novel L-lysine derivatives hybridized a zinc binding group as HDAC inhibitors. No matter their weaker potency compared to that of SAHA against HDAC and cancer cell lines, the study is interesting and of novelty, the inhibitory effect of compound 18 on HDAC and cancer cell lines also confirmed the rationality of the design. However, a few issues still should be addressed before the manuscript can be accepted for publication.

1. The expected proton and carbon number of most compounds differ greatly from those found in the ¹H- and ¹³C-NMR spectra. In addition, there are some unreasonable peaks such as 12.85 ppm of compound 11 and 7.89 ppm of compound 18b carbon peak.
2. What is the purity of the test compound?
3. The structures of 14b, 14c, and 14n on Pages S4, S5, and S8 need to be redrawn.
4. The SD values of the IC₅₀ of 11, 18b, 18c, and SAHA should be added to Data in Table 2.
5. Page 2, left column, lines 47, 52, and 54, change "treating trifluoroacetic acid with compound 7", "the treatment of hydroxyl amine with compound 9", "Treatment of hydroxyl amine with compound 10" to "treating compound 7 with trifluoroacetic acid", "the treatment of compound 9 with hydroxyl amine", "Treatment of compound 10 with hydroxyl amine".
6. On Page 3, left column, line 8, replace "the treatment of hydroxyl amine with intermediates

- 16b-c" with "the treatment of intermediates 16b-c with hydroxyl amine".
7. Page 3, left column, line 5, the sentence "which was under the ester.....give compounds 16a-c" need to be rewritten.
 8. Page 2, left column, line 48, "Followed by...." is incorrect.
 9. "lysine analogs" is incorrect and should be "lysine derivatives".
 10. Move some of the conclusions into the introduction to make the conclusions clear and concise.

Decision letter (RSOS-190338.R0)

29-Mar-2019

Dear Professor Bin:

Title: N ϵ -acetyl-lysine analogues with Zinc binding groups as novel HDAC inhibitors
Manuscript ID: RSOS-190338

The editor assigned to your manuscript has now received comments from reviewers. We would like you to revise your paper in accordance with the referee and Subject Editor suggestions which can be found below (not including confidential reports to the Editor). Please note this decision does not guarantee eventual acceptance.

Please submit your revised paper before 21-Apr-2019. Please note that the revision deadline will expire at 00.00am on this date. If we do not hear from you within this time then it will be assumed that the paper has been withdrawn. In exceptional circumstances, extensions may be possible if agreed with the Editorial Office in advance. We do not allow multiple rounds of revision so we urge you to make every effort to fully address all of the comments at this stage. If deemed necessary by the Editors, your manuscript will be sent back to one or more of the original reviewers for assessment. If the original reviewers are not available we may invite new reviewers.

On behalf of the Subject Editor Professor Anthony Stace and the Associate Editor Dr Andrew Harned.

RSC Associate Editor:

Comments to the Author:

The reviewers expressed some interest in this work and agreed that it looked like a promising approach for HDAC inhibition. However, they do raise several valid concerns (particularly with respect to compound identification and purity) that should be addressed by the authors.

RSC Subject Editor:

Comments to the Author:

(There are no comments.)

Reviewers' Comments to Author:

Reviewer: 1

Comments to the Author(s)

HDAC is an important epigenetic target in drug discovery and development. The authors developed N ϵ -acetyl-lysine analogues with Zinc binding groups as HDAC inhibitors. The best compound 18c had an IC₅₀ value of 500 nM for HDACs. Besides, WB and MTS assay were used to evaluate the antitumor activity of all synthetic compounds. This manuscript has provided an effective strategy for developing novel HDAC inhibitors.

Minor comments:

- 1) the values of the anti-proliferative assay in Fig. 4 should be provided based on dose-dependent curve-fitting.
- 2) in order to evaluate 18c toxicity, the authors need to determine the IC₅₀s of 18c and SAHA against both tumor cells and normal cells, then calculate and compare their selective index.

Reviewer: 2

Comments to the Author(s)

This manuscript described the design and synthesis of HDAC inhibitors by hybridizing acetyl-lysine and zinc binding group. Their enzymatic and cellular activities were also characterized. The most promising compound 18c had reasonable inhibition against HDACs. Interestingly, 18c demonstrate the comparable antiproliferative activities against several cancer cell lines but the less toxicity for normal cells than that of SAHA. Therefore, it is worthy of eventual publication in Royal Society Open Science if the authors can address the following concerns:

1. Should add some discussion about whether the synthesized compounds intend to be pan- or specific- HDAC inhibitors.

2. The cellular data is just described for the acetylation level of tubulin which is only relevant for HDAC6. HDAC1-3 are involved in acetylation of H3, which is strongly suggested to be present in the figure.
3. The authors should give dose-dependent curves in Fig. 4.
4. The authors should check English and careless typos in the text, and the format of the structural formula in SI.

Reviewer: 3

Comments to the Author(s)

N ϵ -Acetyl-lysine analogs with Zinc binding groups as novel HDAC inhibitors
Fang Wang, et al

The manuscript describes the design, synthesis, and evaluation of a series of novel L-lysine derivatives hybridized a zinc binding group as HDAC inhibitors. No matter their weaker potency compared to that of SAHA against HDAC and cancer cell lines, the study is interesting and of novelty, the inhibitory effect of compound 18 on HDAC and cancer cell lines also confirmed the rationality of the design. However, a few issues still should be addressed before the manuscript can be accepted for publication.

1. The expected proton and carbon number of most compounds differ greatly from those found in the ¹H- and ¹³C-NMR spectra. In addition, there are some unreasonable peaks such as 12.85 ppm of compound 11 and 7.89 ppm of compound 18b carbon peak.
2. What is the purity of the test compound?
3. The structures of 14b, 14c, and 14n on Pages S4, S5, and S8 need to be redrawn.
4. The SD values of the IC₅₀ of 11, 18b, 18c, and SAHA should be added to Data in Table 2.
5. Page 2, left column, lines 47, 52, and 54, change "treating trifluoroacetic acid with compound 7", "the treatment of hydroxyl amine with compound 9", "Treatment of hydroxyl amine with compound 10" to "treating compound 7 with trifluoroacetic acid", "the treatment of compound 9 with hydroxyl amine", "Treatment of compound 10 with hydroxyl amine".
6. On Page 3, left column, line 8, replace "the treatment of hydroxyl amine with intermediates 16b-c" with "the treatment of intermediates 16b-c with hydroxyl amine".
7. Page 3, left column, line 5, the sentence "which was under the ester.....give compounds 16a-c" need to be rewritten.
8. Page 2, left column, line 48, "Followed by...." is incorrect.
9. "lysine analogs" is incorrect and should be "lysine derivatives".
10. Move some of the conclusions into the introduction to make the conclusions clear and concise.

Author's Response to Decision Letter for (RSOS-190338.R0)

See Appendix A.

Decision letter (RSOS-190338.R1)

15-Apr-2019

Dear Professor Bin:

Title: N ϵ -acetyl-lysine analogues with Zinc binding groups as novel HDAC inhibitors
Manuscript ID: RSOS-190338.R1

Thank you for submitting the above manuscript to Royal Society Open Science. On behalf of the Editors and the Royal Society of Chemistry, I am pleased to inform you that your manuscript will be accepted for publication in Royal Society Open Science subject to minor revision in accordance with the referee suggestions. Please find the reviewers' comments at the end of this email.

The reviewers and handling editors have recommended publication, but also suggest some minor revisions to your manuscript. Therefore, I invite you to respond to the comments and revise your manuscript.

Please also include the following statements alongside the other end statements. As we cannot publish your manuscript without these end statements included, if you feel that a given heading is not relevant to your paper, please nevertheless include the heading and explicitly state that it is not relevant to your work. We have included a screenshot example of the end statements for reference.

- Ethics statement

Please clarify whether you received ethical approval from a local ethics committee to carry out your study. If so please include details of this, including the name of the committee that gave consent in a Research Ethics section after your main text. Please also clarify whether you received informed consent for the participants to participate in the study and state this in your Research Ethics section.

OR

Please clarify whether you obtained the necessary licences and approvals from your institutional animal ethics committee before conducting your research. Please provide details of these licences and approvals in an Animal Ethics section after your main text.

OR

Please clarify whether you obtained the appropriate permissions and licences to conduct the fieldwork detailed in your study. Please provide details of these in your methods section.

- Data accessibility

It is a condition of publication that you make available the data and research materials supporting the results in the article. Datasets should be deposited in an appropriate publicly available repository and details of the associated accession number, link or DOI to the datasets must be included in the Data Accessibility section of the article (<http://royalsocietypublishing.org/instructions-authors#question17>). Reference(s) to datasets should also be included in the reference list of the article with DOIs (where available).

Please include a Data Availability section after your main text stating where supporting data are available from, or where they will be made available should your article be accepted for publication.

If you wish to submit your supporting data or code to Dryad (<http://datadryad.org/>), or modify your current submission to dryad, please use the following link:
<http://datadryad.org/submit?journalID=RSOS&manu=RSOS-190338.R1>

- **Competing interests**

Please include a Competing Interests section after your main text declaring any financial or non-financial competing interests. If you have no competing interests please state 'I/we have no competing interests.

- **Authors' contributions**

Please include an Authors' Contributions section at the end of your main text detailing the contribution of each author. All authors should have read and approved the manuscript before submission and this should be stated in the Authors' Contributions section.

The list of Authors should meet all of the following criteria; 1) substantial contributions to conception and design, or acquisition of data, or analysis and interpretation of data; 2) drafting the article or revising it critically for important intellectual content; and 3) final approval of the version to be published.

- **Acknowledgements**

- **Funding statement**

Please include a funding section after your main text which lists the source of funding for each author.

Because the schedule for publication is very tight, it is a condition of publication that you submit the revised version of your manuscript before 24-Apr-2019. Please note that the revision deadline will expire at 00.00am on this date. If you do not think you will be able to meet this date please let me know immediately.

Best wishes,
Dr Laura Smith
Publishing Editor, Journals

On behalf of the Subject Editor Professor Anthony Stace and the Associate Editor Dr Andrew Harned.

RSC Associate Editor
Comments to the Author:

The authors have done a good job responding to the comments and concerns raised by the previous review. However, there are still a few relatively minor items that need to be addressed before I am comfortable recommending final acceptance.

- (1) Overall the grammar of the manuscript could still be improved. The authors are strongly encouraged to seek assistance in this area to that their message is not lost during final editing. In particular, the paragraph at the top right of Page 2 needs to be edited for clarity and message.
- (2) Please include copies of ^1H and ^{13}C NMR spectra in the supporting information.

Author's Response to Decision Letter for (RSOS-190338.R1)

See Appendix B.

Decision letter (RSOS-190338.R2)

30-Apr-2019

Dear Professor Bin:

Title: N ϵ -acetyl-lysine analogues with Zinc binding groups as novel HDAC inhibitors
Manuscript ID: RSOS-190338.R2

It is a pleasure to accept your manuscript in its current form for publication in Royal Society Open Science. The chemistry content of Royal Society Open Science is published in collaboration with the Royal Society of Chemistry.

On behalf of the Subject Editor Professor Anthony Stace and the Associate Editor Dr Andrew Harned.

RSC Associate Editor
Comments to the Author:
Publication is recommended at this time.

Reviewer(s)' Comments to Author:

Appendix A

Bin He
Professor
College of Pharmacy
Guiyang Medical University
Guiyang, Guizhou 550004 China

Tel: +86-13765113985
Fax: +86-851-6908218
E-mail: binhe@gmc.edu.cn

April 4, 2019

Dear Editor,

We are submitting the revision of our manuscript **RSOS-190338** entitled “N ϵ -acetyl-lysine derivatives with Zinc binding groups as novel HDAC inhibitors”. We highly appreciate the editor’s and reviewers’ comments as they are important for the improved quality of this manuscript. Our responses/changes to the reviewers’ comments are detailed below.

Responses to Editor’s Comments

According to the Reviewers’ comments, we have done the point-to-point response shown as below. Those corrections and modifications in text and SI have been highlighted by blue color.

Responses to Reviewer 1’s Comments

Reviewer 1 is very positive about the manuscript. We thank the reviewer for the helpful comments and careful reading of the manuscript. Our changes and/or responses are detailed below.

Comment 1: “the values of the anti-proliferative assay in Fig. 4 should be provided based on dose-dependent curve-fitting.”

Response: *The dose-dependent curve-fitting of anti-proliferation experiment has been replaced in Figure 4 showing as below:*

Fig 4

Comment 2: “in order to evaluate **18c** toxicity, the authors need to determine the IC₅₀s of **18c** and SAHA against both tumor cells and normal cells, then calculate and compare their selective index.”

Response: Upon the reviewer’s request, we have done the IC₅₀s of **18c** and SAHA against both tumor cells and normal cells, then calculate and compare their selective index, which is shown in the Table. S2 of Supporting Information.

	K562	A549	HepG2	HEK293	selectivity index (SI)
SAHA	4.58±0.17	17.81±1.25	4.23±0.63	6.09±0.84	0.34~1.43
18c	41.18±1.73	134.10±2.13	158.40±2.20	>500	3.16~12.14
16c	>500	>500	>500	>500	/

Responses to Reviewer 2’s comments

Reviewer 2 is also very positive about the manuscript. We thank the reviewer for the helpful comments and careful reading of the manuscript. Our changes and/or responses are detailed below.

Comment 1: “Should add some discussion about whether the synthesized compounds intend to be pan- or specific- HDAC inhibitors.”

Response: By using the substrates of corresponding subtypes of HDAC, we tested the inhibitory activity at a final concentration 1μM of compounds **18c** and SAHA, and the results showed that compound **18c** like SAHA is a pan-HDAC inhibitors, which is shown in the Table. S1 of Supporting Information.

Inhibitor*	HDAC Inhibition (%)		
	HDAC I	HDAC IIa	HDAC8
SAHA	87.05±2.193	15.51±3.084	41.19±0.180
18c	61.06±1.064	16.65±2.690	38.27±1.215

*1μM

Comment 2: “ The cellular data is just described for the acetylation level of tubulin which is only relevant for HDAC6. HDAC1-3 are involved in acetylation of H3, which is strongly suggested to be present in the figure. ?”

Response: Considering H3 and Tubulin are substrates commonly used for the cellular study of HDAC inhibitors, we do have added their acetylation level of tubulin and H3 shown in the upper panel for each cell lines in Figure 3.

Comment 3: “ The authors should give dose-dependent curves in Fig. 4.”

Response: The dose-dependent curve-fitting of anti-proliferation experiment has been replaced in Figure 4 showing as below:

Comment 4: “ The authors should check English and careless typos in the text, and the format of the structural formula in SI. ”

Response: *Thanks for the careful reading, we have double-checked although the text and SI and corrected all typos.*

Reponses to Reviewer 3’s comments

Reviewer 3 is in general enthusiastic about the manuscript with raising several concerns and pointing out some typos. We thank the reviewer for the helpful comments and careful reading of the manuscript. Our changes and/or responses are detailed below.

Comment 1: “The expected proton and carbon number of most compounds differ greatly from those found in the ¹H- and ¹³C-NMR spectra. there are some unreasonable peaks such as 12.85 ppm of compound **11** and 7.89 ppm of compound **18b** carbon peak.”

Response: *Thanks for the careful reading, we have double-checked the proton and carbon numbers. The reason is that some compounds contain some solvents or possess active protons (-COOH, -CONH-). Therefore, we have marked the identical peaks of solvents, water or those active protons. Additionally, some unreasonable peaks such as 12.85 ppm of compound **11** and 7.89 ppm of compound **18b** carbon peak are those peaks of petroleum ether. We have solved these problems.*

Comment 2: “What is the purity of the test compound?”

Response: *We are sorry for missing the purity. The purity of all tested compounds was over 95% by HPLC. We have added the corresponding description in SI.*

Comment 3: “The structures of **14b**, **14c**, and **14n** on Pages S4, S5, and S8 need to be redrawn.”

Response: *Thanks for the careful reading, we have corrected this typo with redrawing the structures of **14b**, **14c**, and **14n** on Pages S4, S5, and S8.*

Comment 4: “ The SD values of the IC₅₀ of **11**, **18b**, **18c**, and **SAHA** should be added to Data in Table 2.”

Response: *Thanks for this kind remind, we have added the SD values of the IC₅₀ of **11**, **18b** and **18c** into Table 2.*

Comment 5: “Page 2, left column, lines 47, 52, and 54, change “treating trifluoroacetic acid with compound 7”, “the treatment of hydroxyl amine with compound 9”, “Treatment of hydroxyl amine with compound 10” to “treating compound 7 with trifluoroacetic acid”, “the treatment of compound 9 with hydroxyl amine”, “Treatment of compound 10 with hydroxyl amine”. ”

Response: *Thanks for the careful reading, we have corrected these mistakes.*

Comment 6: “On Page 3, left column, line 8, replace “the treatment of hydroxyl amine with intermediates 16b-c” with “the treatment of intermediates 16b-c with hydroxyl amine”.

Response: *Thanks for the careful reading, we have corrected this typo.*

Comment 7: “Page 3, left column, line 5, the sentence “which was under the ester.....give compounds 16a-c” need to be rewritten. ”

Response: *Thanks for the careful reading, we have rewritten the sentence as “which was hydrolyzed to give compounds 16a-c.”*

Comment 8: “Page 2, left column, line 48, “Followed by....” is incorrect. ”

Response: *Thanks for the careful reading, “Followed by....” has been corrected to “ After”.*

Comment 9: “lysine analogs” is incorrect and should be “lysine derivatives”.”

Response: *Thanks for the careful reading, I have corrected this typo.*

Comment 10: “Move some of the conclusions into the introduction to make the conclusions clear and concise.”

Response: *Thanks for the kind suggestion, we have moved and combined the first paragraph to the introduction to make the conclusions clear and concise.*

We thank all the reviewers again for careful reading of the manuscript and for their helpful comments. We hope the revised manuscript have addressed all their concerns.

Sincerely,

Bin He

Bin He

Appendix B

Bin He
Professor
School of Pharmacy
Guizhou Medical University
Guizhou, Guizhou 550004 China

Tel: +86-13765113985
Fax: +86-851-6908218
E-mail: binhe@gmc.edu.cn

April 18, 2019

Dear Editor,

We are submitting the revision of our manuscript **RSOS-190338** entitled “N ϵ -acetyl-lysine derivatives with Zinc binding groups as novel HDAC inhibitors”. We highly appreciate the editor’s and reviewers’ comments as they are important for the improved quality of this manuscript. Our responses/changes to the reviewers’ comments are detailed below.

Responses to Editor’s Comments

Comments: “Please also include the following statements alongside the other end statements. As we cannot publish your manuscript without these end statements included, if you feel that a given heading is not relevant to your paper, please nevertheless include the heading and explicitly state that it is not relevant to your work. We have included a screenshot example of the end statements for reference.”

Response: *We are grateful for the kind reminder. We have included all the required statements alongside the other end statements.*

Responses to Reviewer’s Comments

The reviewer is very positive about the manuscript. We thank the reviewer for the helpful comments and careful reading of the manuscript. According to the Reviewers’ comments, we have done the point-to-point response shown as below. Those corrections and modifications in text and SI have been highlighted by blue color. Our changes and/or responses are detailed below.

Comment 1: “Overall the grammar of the manuscript could still be improved. The authors are strongly encouraged to seek assistance in this area to that their message is not lost during final editing. In particular, the paragraph at the top right of Page2 needs to be edited for clarity and message”

Response: Thank the reviewer’s careful reading. We have improved the grammar through the manuscript. Especially, *for the paragraph at the top right of Page2, “Given HDAC is one of attractive drug targets found in recent years, …… have led to many successful examples.²⁷⁻³⁰” has been edited to “Although considerable progress has been made in the development of HDAC inhibitors, clinically used HDAC inhibitors still have some side effects, such as excessive toxicities, instability and off-target effects.¹⁹⁻²³ Therefore, the development of novel HDAC inhibitors are continuously needed to avoid side effects and improve pharmacological and pharmacokinetic properties.²⁴⁻²⁶ At present, there are few*

HDAC inhibitors designed from Nε-acetyl lysine (HDAC substrate) while the designs of sirtuin inhibitors by mimicking Nε-acyl lysine (sirtuin substrate) have led to many successful examples.²⁷⁻³⁰

Comment 2: “ Please include copies of ¹H and ¹³C NMR spectra in the supporting information. ”

Response: *We have placed the copies of ¹H and ¹³C NMR spectra of all the compounds into the supporting information (please see pages S16 - S47).*

Other modifications

To make the interpretation clearer in Fig. 4, we have modified the “Log10C(Conc)” to “Log10 [Cmpd(uM)]”, and added the corresponding statement of “(Cmpd: tested compound **16c**, **18c** or **SAHA**)” in Fig. 4.

We thank all the reviewers again for careful reading of the manuscript and for their helpful comments. We hope the revised manuscript have addressed all their concerns.

Sincerely,

Bin He

Bin He